# Effects of Damage to the Integrity of the Left Dual-Stream Frontotemporal Network Mediated by the Arcuate Fasciculus and Uncinate Fasciculus on Acute/Subacute Post-Stroke Aphasia

**DOI:** 10.3390/brainsci13091324

**Published:** 2023-09-14

**Authors:** Qiwei Yu, Yuer Jiang, Yan Sun, Xiaowen Ju, Tianfen Ye, Na Liu, Surong Qian, Kefu Liu

**Affiliations:** 1Department of Rehabilitation Medicine, The Affiliated Suzhou Hospital of Nanjing Medical University, Suzhou 215008, China; 20214132128@stu.suda.edu.cn (Q.Y.); yuerjiang@yeah.net (Y.J.); xiaowenju0207@163.com (X.J.); ytf771009@163.com (T.Y.); liuna20220805@126.com (N.L.); qiansr65506@163.com (S.Q.); 2Department of Radiology, The Affiliated Suzhou Hospital of Nanjing Medical University, Suzhou 215008, China; zixiaoshanshi2022@163.com

**Keywords:** dual-stream pathways, arcuate fasciculus, uncinate fasciculus, post-stroke aphasia, diffusion tensor imaging

## Abstract

(1) Background: To investigate the correlation between the integrity of the left dual-stream frontotemporal network mediated by the arcuate fasciculus (AF) and uncinate fasciculus (UF), and acute/subacute post-stroke aphasia (PSA). (2) Methods: Thirty-six patients were recruited and received both a language assessment and a diffusion tensor imaging (DTI) scan. Correlations between diffusion indices in the bilateral LSAF/UF and language performance assessment were analyzed with correlation analyses. Multiple linear regression analysis was also implemented to investigate the effects of the integrity of the left LSAF/UF on language performance. (3) Results: Correlation analyses showed that the diffusion indices, including mean fractional anisotropy (FA) values and the fiber number of the left LSAF rather than the left UF was significantly positively associated with language domain scores (*p* < 0.05). Multiple linear regression analysis revealed an independent and positive association between the mean FA value of the left LSAF and the percentage score of language subsets. In addition, no interaction effect of the integrity of the left LSAF and UF on language performance was found (*p* > 0.05). (4) Conclusions: The integrity of the left LSAF, but not the UF, might play important roles in supporting residual language ability in individuals with acute/subacute PSA; simultaneous disruption of the dual-stream frontotemporal network mediated by the left LSAF and UF would not result in more severe aphasia than damage to either pathway alone.

## 1. Introduction

The contemporary model of language organization has highlighted the important linguistic contribution of the left frontotemporal network. Modern views emphasize that language facility relies on functionally coordinated interactions of multiple brain regions located in the left inferior frontal, anterior, posterior, and inferior temporal lobes that constitute a left-lateralized frontotemporal network [1]. Neuroimaging studies have revealed that the functional interactions between these frontal and temporal areas critically depend on the efficient transmission of information, and that this information flow is subserved by structural (sub)cortico-cortical connections via multiple white matter fiber pathways [2,3,4].

Hickok and Poeppel [5,6] proposed a dual stream model of language processing in the brain, including a dorsal and a ventral stream, which was one of the most influential contemporary neuropsychological models of speech and language organization. As described in this model, the dorsal stream is organized across the frontal speech areas and a region located at the temporal-parietal junction—which is responsible for the processing of auditory-to-articulation information—while the ventral stream roots in the temporal lobes and supports the organization of auditory-to-meaning information [7]. In a study evaluating the relationship between language deficits and stroke lesion locations, Fridriksson et al. [8] mapped the grey matter localization of the dorsal and ventral streams. Consistent with the viewpoints of Hickok and Poeppel, Fridriksson et al. [7,8] suggested that the dorsal stream involved some cortical areas, including the pars opercularis, pars triangularis, and pre-and postcentral regions, mainly located in the frontoparietal lobes, while the ventral stream mainly consisted of the lateral temporal lobe, extending into the posterior-inferior frontal gyrus pars orbitalis.

In terms of white matter, fiber pathways formed the anatomical basis of the human language connectome, and several major fiber tracts constitute the dual-pathway language model: the dorsal pathway mainly consists of the arcuate fasciculus (AF) and the superior longitudinal fasciculus (SLF); and the ventral pathway is composed of the inferior longitudinal fasciculus (ILF), uncinate fasciculus (UF), the inferior frontal-occipital fasciculus (IFOF), and the middle longitudinal fasciculus (MdLF) [9]. The AF is the major dorsal fiber pathway connecting the frontal lobe with the temporal lobe, and it is currently one of the most investigated neural tracts involved in language processing [10,11]. The SLF was suggested to be a complex fiber system consisting of a least three subcomponents connecting the frontal areas with the angular and supramarginal gyrus, and the ILF connects the anterior temporal region to the posterior occipitotemporal regions. The UF is a hook-shaped tract that originates from the anterior temporal lobe, parahippocampal gyrus, and amygdala, and terminates in the basal and lateral portions of the frontal lobe (i.e., the pars orbitalis and triangularis of the IFG). The IFOF is the longest associative pathway connecting the frontal lobe to the temporobasal areas, superior parietal lobe, and occipital cortex. To date, the MdLF is one of the least studied and poorly understood association tracts connecting the temporal and parietal regions [2,12,13]. Although growing literature has reported the involvement of these fiber tracts in physiological and pathological language processing, the linguistic roles of white matter fiber tracts remain rather unclear and have been explored for dedicates.

Given the anatomical trajectory of the dual stream pathways of language organization in the brain, the dorsal stream is suggested to both support mapping sound onto articulatory-based representations and map acoustic speech signals to frontal lobe articulatory networks, while the ventral stream likely contributes to mapping sound onto meaning and processes speech signals for comprehension, as proposed by Hickok and Poeppel [5,6]. Likewise, according to neurocomputational models of language organization, Ueno et al. [14,15] and Bornkessel-Schlesewsky et al. [16] proposed that, to some extent, both the dorsal and ventral streams contribute to each language task, arguing for a “graded division of labor”—the dorsal pathway being involved in time-variant sound-to-motor mappings, and the ventral pathway being responsible for time-invariant sound-to-meaning mappings. These findings supported the differential linguistic contributions of the dorsal and ventral pathways.

However, there is evidence to support a close interaction between the dorsal and ventral streams in the successful execution of language facility [13,17]. For instance, Weiller et al. [18] and Rolheiser et al. [19] advocated a strong interaction and synergistic system rather than a segregated one between both the dorsal and the ventral pathways. Furthermore, based on the viewpoint of interaction and synergistic systems, Zavanone et al. [20] proposed that simultaneous disruption of the dorsal and ventral pathways might hamper “compensatory” utilization of one pathway after damage to the other one. However, how the two processing pathways interact is rather unclear [17]. Accordingly, whether these dual-stream pathways support language processing independently or interactively remains a matter of debate, and it is an important field for exploration.

In the current work, we aimed to investigate the effects of disruption of the integrity of the left frontotemporal network on acute/subacute post-stroke aphasia (PSA). We focused on investigating its correlation with the integrity of the white matter fiber pathways rather than the activation of cortical regions. Based on the findings of previous literature, we selected the AF as the targeted neural tract of the dorsal stream pathways due to its direct connection to the frontal and temporal lobe; likewise, we selected the UF as the targeted neural tract of the ventral stream pathways. Specifically, the anatomical details of AF were taken into account in this work. In 2005, Catani et al. [11] outlined a three-segment anatomical model of the bilateral AFs. According to this model, the AF could be subdivided into one long projection fiber (i.e., a long segment, LSAF) and two short projection fibers (i.e., an anterior and a posterior segment). The LSAF suggested that it directly connects Wernicke’s area to Broca’s area, the anterior segment (ASAF) linked Broca’s area with Geschwind’s area (i.e., the inferior parietal lobule), while the posterior segment (PSAF) projected from Geschwind’s area to Wernicke’s area, as seen in Figure 1. It was reported that every AF segment might differently contribute to language functions [21,22,23]. To better understand the language functions of the AF, it was recently suggested to take its anatomical details into account and not roughly treat it as a unifunctional entity when exploring the linguistic role of the AF [21,22]. Hence, we evaluated the integrity of the long AF segment (the LSAF) rather than the whole AF fiber bundle.

We assumed that both the AF and UF were involved in language processing and hypothesized that disruption of the dual-stream frontotemporal network mediated by the left LSAF and UF would have an important impact on language abilities in individuals with acute/subacute PSA, and that simultaneous disconnection of the left LSAF and UF would affect language ability significantly more than damage to either pathway alone.

## 2. Materials and Methods

### 2.1. Participants

Thirty-six participants (27 males/9 females, age range: 25–78 years, mean age: 53.31 ± 12.37 years; mean duration of education: 10.78 ± 3.48 years; mean time since stroke: 36.56 ± 28.25 days) with acute/subacute PSA were recruited according to the following inclusion criteria: first-ever left hemispheric stroke (i.e., infarction or hemorrhage) with normal consciousness, time post-stroke less than 90 days, age between 18–80 years, right-handedness and native Chinese speakers, the consistent presence of various types of aphasia (aphasia quotient value < 93.8), collaborating to complete a language faculties assessment and magnetic resonance imaging (MRI) scan. Those who met one of the following criteria would be excluded: history of stroke, right or bi-hemispheric stroke, duration of stroke more than 90 days, left-handedness, concomitant neuropsychiatric diseases, other aphasic syndromes such as primary progress aphasia, and being unable to finish language faculties assessment or MRI scan.

This study protocol was approved by the Institutional Review Board of our hospital, and informed consent was obtained from one family member of every participant.

### 2.2. Language Assessment

All patients underwent language assessment with the Chinese version of the Western Aphasia Test (WAB) after admission [24,25]. The language domains, including spontaneous speech (i.e., information content and verbal fluency), auditory verbal comprehension (i.e., yes/no questions, word recognition, and sequential commands), repetition (i.e., word, phrase, and sentence repetition), and naming (i.e., object naming, word fluency, sentence completion, and responsive speech) were used to assess the patients’ language ability, and the percentage score of each sub-item was calculated for analysis according to an equation: percentage score = (patient’s score/total score of each sub-item) × 100%. A high percentage score of each sub-item usually represents relatively good language performance. In addition, the global aphasia severity of every patient was determined based on the aphasia quotient (AQ) value of the WAB. According to the grading criteria [26], the aphasia severity was classified into four levels: very severe (AQ, 0–25), severe (AQ, 26–50), moderate (AQ, 51–75), and mild (AQ, ≥76). The language assessment of all participants was finished by one speech-language therapist.

### 2.3. Image Data Acquisition and Pre-Processing

Image data were obtained using a 3.0 Tesla Siemens Skyra scanner with a standard radio-frequency head coil. High-resolution three-dimensional T1-weighted structural image data of the whole brain tissue were acquired using a magnetization-prepared rapid gradient-echo (MPRAGE) protocol according to the following parameters: repetition time (TR), 2300 ms; echo time (TE), 2.98 ms; inverse time, 900 ms; flip angle, 9°; number of averages, 1; the field of view (FOV), 256 × 248 mm^2^; slice thickness/slice spacing, 1.1 mm/0 mm; sagittal slices, 176; total acquisition time, 5.2 min. The diffusion tensor imaging (DTI) data were obtained with a single-shot echo planar imaging (EPI) sequence in 49 contiguous axial slices parallel to the anterior-posterior (AC-PC) plane following the parameters: TR, 7200 ms; TE, 104 ms; flip angle, 90°; FOV, 896 × 896 mm^2^; acquisition matrix, 96 × 96 mm^2^; reconstructed matrix, 128 × 128 matrices; two b-values, 0 and 1000 s/mm^2^; diffusion sensitive gradient directions, 64; slice thickness/slice spacing, 2.5 mm/2.5 mm; total acquisition time, 8.3 min. All image data acquisition was completed by one radiologist.

All original T1-weighted and diffusion image data were pre-processed with the FMRIB Software Library (FSL version 5.0.9 software package, http://fsl.fmrib.ox.ac.uk/fsl/fslwiki/FSL, accessed on 10 October 2022) in four steps [27]. In the first step, all DICOM files were converted into NIfTI format files by means of the “dcm2niigui” tool. In the second step, head motion and eddy currents were corrected with the “eddy_correct” function. In the third step, skull-stripping and removal of non-brain tissue was performed using the “BET v2.1” function. In the fourth step, the diffusion tensor model was linearly fitted to the diffusion-weighted images (DWI) by applying the “DTIFIT” tool. Then, fractional anisotropy (FA) images were generated for each patient. After that, both the processed T1-weighted and FA images of every patient were spatially normalized into the standard MNI152 atlas space by using the “FLIRT” tool within FSL.

Lesion maps were manually drawn slice by slice on the normalized high-resolution T1-weighted structural images using MRIcroGL (https://www.mccauslandcenter.sc.edu/mricrogl/, accessed on 16 October 2022), referring to T2-weighted images and DWI images.

### 2.4. DTI Tractography

The Diffusion Toolkit (version 0.6.4.1, http://trackvis.org/ (accessed on 25 October 2022), Martinos Center for Biomedical Imaging, Massachusetts General Hospital) was used for the reconstruction of whole-brain diffusion tensors, and TrackVis (version 0.6.1, http://trackvis.org/ (accessed on 25 October 2022), Martinos Center for Biomedical Imaging, Massachusetts General Hospital) software was applied to manually delineate the region of interest (ROI) and the visualization fiber tracking on the normalized FA images [28]. Multiple ROIs approaches were applied to accomplish the reconstruction of the target fiber tracts based on a deterministic fiber-tracking algorithm (fiber assignment by continuous tracking, namely FACT algorithm). Fiber tracking was initiated with a minimum FA value at 0.20 and an angle threshold of 35°.

To reconstruct the LSAF, a combination of three ROIs approaches were used to accomplish virtual dissection of the bilateral AFs: a frontal ROI (ROI 1) placed on the coronal slice at the entrance to the frontal lobe (anterior to the central sulcus), a temporal ROI (ROI 2) delineated in the axial slice at the entrance to the temporal lobe (below the Sylvian fissure), and a parietal ROI (ROI 3) placed tangentially to the inferior parietal cortex [21,22]. The streamlines going through both ROI 1 and ROI 3 but not ROI 2 were defined as the LSAF [22,23].

To reconstruct the UF, the first ROI was placed on the temporal stem and anterior floor of the external/extrema capsules, and the second ROI was located in the anterior white matter of the temporal lobe [29,30].

### 2.5. Evaluation of the White Matter Pathways

Diffusion indices including the mean FA value and fiber number were used to quantitatively evaluate the structural integrity of target neural tracts. To identify the anatomical trajectory and spatial location, the reconstructed LSAF, UF, and stroke lesions were visualized on a standard T1-weighted image atlas (the MNI152 atlas).

### 2.6. Statistical Analyses

All statistical analyses were performed with the IBM SPSS software package (version 25.0). The Shapiro–Wilk test was applied to evaluate the normality of the data. The Pearson correlation analyses were used for parametric variables, while the Spearman correlation analyses were used for non-parametric variables. In addition, multiple linear regression analysis was also implemented to investigate the associations between language performance and the integrity of the LSAF/UF. Demographic and stroke-related variables, including sex, age, educational level, time post of stroke, and lesion volume, were selected as confounding variables. The variance inflation factor (VIF, >5) was used to diagnose multi-collinearity between variables in the multiple linear regression models. A *p* value less than 0.05 was considered statistically significant.

## 3. Results

### 3.1. General Demographic and Clinical Characteristics

All thirty-six patients completed the language abilities assessment and MRI scan. All participants were native Chinese speakers. Twenty-one patients suffered from ischemic strokes (infarction), and others suffered from intracerebral hemorrhages (ICH). There were twenty-four patients with non-fluent aphasia, including Broca’s aphasia (7 cases), transcortical motor aphasia (TMA, 3 cases), mixed transcortical aphasia (MTA, 1 case), and global aphasia (13 cases), and twelve patients with fluent aphasia, including Wernicke’s aphasia (3 cases), anomic aphasia (4 cases), conduction aphasia (3 cases), and transcortical sensory aphasia (TSA, 2 cases). The aphasia severity level of all the patients ranged from mild to very severe, according to the criteria [26]. The general demographic and clinical data of all participants are presented in Table 1.

### 3.2. Correlation Analyses between Language Assessment and MRI Measures

First, we analyzed the correlations between language performance assessment and demographic and stroke-related variables. Pearson correlation analyses were applied to analyze the correlation between age, AQ, and scores of spontaneous speech, respectively; and Spearman correlation analyses were used to estimate the correlations between education, time post of stroke, and language performance, including comprehension, repetition, naming, and speech fluency. There was no significant correlation between both demographic and stroke-related variables and language performance (*p* > 0.05).

Then, we analyzed the correlations between language performance assessment and MRI metrics including lesion volume and diffusion indices of the bilateral LSAF/UF with Spearman correlation analyses due to non-normally distributed variables. The results showed that lesion volume had a moderately negative relation to AQ (ρ = −0.359, *p* = 0.032) and the score of naming (ρ = −0.351, *p* = 0.036); the mean FA values and fiber number of the left LSAF had moderately to strongly positive correlations with scores of any language subsets, while the mean FA values of the right LSAF/UF had moderately positive associations with scores of some language subsets. However, no significant correlation was found between diffusion indices of the left UF and language measures, as seen in Table 2. We further evaluated the partial correlation coefficients between diffusion indices of the left LSAF and the language domain scores while removing the effects of stroke lesion volume. When lesion volume was set as a controlled variable, partial correlation analyses revealed that the diffusion indices of the left LSAF still had moderately to strongly positive correlations with all language domain scores (*p* < 0.05), as seen in Figure 2.

### 3.3. Multiple Linear Regression Analysis

We further estimated the correlations between the language abilities assessment and diffusion indices of the left LSAF/UF using multiple linear regression analysis. The results revealed an independent and positive association between the mean FA value of the left LSAF and scores of language subsets. The fiber number of the left LSAF was mainly positively related with the AQ value (B = 0.088, standard Beta = 0.636, *p* = 0.036, F = 1.767) and scores of comprehension (B = 0.010, standard Beta = 0.767, *p* = 0.013, F = 1.829). However, the diffusion indices of the left UF were not significantly associated with language performance. No significant correlations were found between the language assessment and any other demographic or clinical metrics (*p* > 0.05). Furthermore, the results showed no interaction effects of the integrity of the left LSAF and UF on language performance (*p* > 0.05). Collinearity analyses indicated no co-linearity among the variables (VIF less than 5 for all multiple linear regression models), as seen in Table 3.

### 3.4. Lesion Overlay Map

The lesion volumes of all participants: range = 1.44–203.50 cm^3^, mean (SD) = 49.70 (44.59) cm^3^. The lesion overlay map demonstrated a heterogeneous distribution of stroke lesion sites among the participants, as seen in Figure 3.

## 4. Discussion

In this work, we investigated the correlation between the integrity of the left dorsal and ventral frontotemporal network mediated by the LSAF and UF, respectively, and the language performance of patients with acute/subacute PSA. Our results suggested the potential importance of the left LSAF but not the UF in supporting residual language ability in individuals with acute/subacute PSA. Meanwhile, no interaction effect of damage to the left LSAF and UF on PSA was found in our patient cohort.

Since Catani and colleagues [11] proposed the three-segment model of the AF anatomy details, the importance of evaluating the linguistic roles of tract segments rather than whole tracts has been recognized, and increasing studies reported distinct language functions of the AF segments [21,22,23]. Catani et al. [11] speculated that the LSAF might be involved in phonological processing (e.g., speech repetition). López-Barroso et al. [31] supported the contribution of the left LSAF to novel word learning, and Gullick et al. [32] reported the role of the LSAF in predicting reading ability.

In studies on PSA, Ivanova et al. [21] found a potential correlation between the micro-structural integrity of the left LSAF, and auditory comprehension and naming ability in individuals with chronic PSA. We recently reported the correlation between the left LSAF and aphasia severity, auditory comprehension, and naming in acute/subacute PSA [22]. In this study, the results of correlation analyses showed moderate to strong positive correlations between the integrity of the left LSAF and language domains, including aphasia severity, comprehension, and naming, of language assessment in populations with acute/subacute PSA, which were further determined with multiple linear regression analysis. Indeed, it was reported that the left LSAF might be involved in the pragmatic integration and higher cognitive function processes of language, as well as in the processing of syntactically complex sentences [33,34]. Thus, disruption of the integrity of the left LSAF could cause deficits in the processing of complex syntactic structures, which results in impairments in the comprehension of noncanonical sentences [35,36,37].

Notably, our results also revealed a significant positive correlation between the integrity of the left LSAF and spontaneous speech, repetition, and fluency. Gajardo-Vidal et al. [38] recently pointed out that damages to both the left ASAF and LSAF were likely to result in long-lasting speech production impairments after a lesion involved in Broca’s area. Kernbach et al. [39] suggested that both the ASAF and LSAF might play roles in predicting the prognosis of speech fluency. However, we could not come to conclusions about the general linguistic roles of the left LSAF in spontaneous speech, repetition, and fluency; based on the findings of previous literature, it was suggested that spontaneous speech disturbances and speech fluency impairment might be mainly associated with damage to the ASAF [40,41,42] while repetition deficits were related to damage to the PSAF [21,23,43,44]. We speculated that the direct connection between the left frontal and temporal lobes, which were important for successful speech production, repetition, and speech fluency, might explain the correlation between LSAF and speech, repetition, and fluency in our study.

It was suggested that the UF might have a dual functionality: one related to a broad range of behaviors, including anxiety, psychopathy, schizophrenia, frontotemporal dementia, and autism spectrum disorder [45,46]; and the other to lexico-semantic processing [33,47]. The association between the UF and language processing may be speculations regarding its anatomical position and cortical terminations [48]. The anterior temporal cortex (especially the temporal pole) formed the caudal termination of the UF and was regarded as a potential ‘hub’ of a distributed semantic memory network [49], while the inferior frontal lobe (especially the orbitofrontal cortex) was suggested to involve encoding and processing names [9,50]. Both the anterior temporal lobes and portions of the frontal lobes have been proposed to encode, store, and retrieve semantic knowledge [51,52]. Therefore, the UF (mainly the left UF) might support broad language domains, including semantic processing [47], fluency and object naming [53,54], speech production deficits [55], and syntactic processes [56].

However, several studies reported no significant correlation between the integrity of the UF and language processing. For example, Breier et al. [57] found that damage to the UF was not associated with deficits in confrontation naming. Van Hees et al. [58] found no structural changes in the bilateral UF following naming treatment in chronic aphasia, and Ivanova et al. [42] suggested no effects of damage to the UF on language processing in chronic PSA. Additionally, the roles of the left AF and UF have been jointly investigated in several studies on PSA. Marchina et al. [59] reported that damage to the AF, but not to the UF, was significantly associated with impairments in spontaneous speech, including speech rate, informativeness, overall efficiency, fluency, and naming skills. Coincidentally, Hope et al. [54] also found that speech fluency and naming accuracy were significantly negatively related to the lesion load of the left AF but not of the UF in chronic PSA. Rosso et al. [60] found that the global severity of chronic aphasia was related to a critical white matter crossroad area overlapping with the left AF and the IFOF, but not with the left UF. Consistent with these previous findings, our results also supported the important influence of damage to the left LSAF rather than the UF on PSA. Differently from these studies, we took into account the sub-components of the left AF and investigated the linguistic involvement of the left LSAF while these prior studies explored the language functions of the whole AF entity.

Based on previous findings, several possible factors should be considered to account for the controversy on the linguistic role of the UF. Firstly, the UF anatomically connects regions located at the ventral and medial frontal lobe (which are not commonly associated with linguistic function) rather than the classic frontal language areas (i.e., Broca’s area/inferior frontal gyrus), to the anterior temporal lobe and surrounding structures [45]. Second, the UF was suggested to comprise multiple sub-components. For example, Ebeling et al. [61] proposed a three-segment anatomical model of the UF, and Hau et al. [62] reported five subcomponents of the UF. Each subcomponent had different asymmetries and distinct functions. However, the linguistic roles of the subcomponents of the UF were rarely reported. Due to the limitation of DTI tractography methodology, we could not subdivide the UF and investigate the correlation between the lateralization of the UF subcomponents and linguistic functions.

Additionally, the dorsal stream (i.e., the AF) was suggested to be more strongly left lateralized while the ventral stream pathways, including the UF, were bilaterally represented [2,6]. Therefore, the left AF is more likely to be destroyed by left hemispheric lesions, while the linguistic roles of the left UF may be more easily compensated by the contralateral UF after left hemisphere lesions [59,63]. Catani et al. [64] reported an extreme left lateralization of the direct long segment in ~60% of the right-handed population, and found that the extreme left lateralization of the direct long segment was significantly associated with worse performance on a complex verbal memory task. They speculated that asymmetry of the direct segment might represent a key anatomical substrate for language lateralization [52,64]. According to the method of Catani et al. (2007), we evaluated the lateralization of the LSAF and UF in intra-individuals with an intact left LSAF (patient ID: 1–17, Table 1) and UF (patient ID: 1–8 and 18–27, Table 1). As a result, we found a leftward distribution of the fiber number index of the LSAF (88.24%, mean ratio = 0.07 ± 0.082) and a bilateral distribution of the fiber number index of the UF (mean ratio = 0.00 ± 0.096). Our results were consistent with the previous reports [29,58,64,65], which support the potential linguistic role of the left AF rather than the UF in acute/subacute PSA; the lateralization property of the LSAF and UF might be one of the explanations for our findings.

Finally, the UF was postulated to act as a subsidiary role with the IFOF [66,67]. Duffau et al. [63,68] suggested that the “semantic ventral stream” might be constituted by two parallel pathways: a direct pathway, including the IFOF, which directly connected the posterior temporal cortex to the inferior frontal lobe and was crucial for semantic processing; and an indirect pathway, consisting of the ILF linking the posterior temporal cortex with the anterior temporal lobe (ATL) and the UF, projecting from the inferior frontal lobe to the ATL, which is not systematically indispensable and could be compensated during stimulation and after surgical resection [69], as seen in Figure 4. In the study of Rosso et al. [60], the global severity of chronic aphasia was found to be related to a combined disconnection both of the left AF and the IFOF instead of the UF. Therefore, they suggested that simultaneous disruption of both the dorsal and ventral streams might provide explanations for the strong correlation with aphasia severity in chronic stroke patients. To elucidate the impacts of the IFOF on the linguistic role of the UF, we attempted to measure the integrity of both the left IFOF and UF (the reconstruction methods of the IFOF referred to in references [29,30]). However, we could not distinguish herein the effects of damage to the left UF and IFOF on language performance due to a high co-occurring injury rate of the UF and IFOF in our patients’ cohort, as shown in Figure 5. Therefore, further exploration of the functional relationship between the UF and IFOF is warranted.

Several limitations should be mentioned in this study. First, the sample of each group subject was relatively small, which limits the generalizability of the results. Second, we did not evaluate another neural tract, namely the extreme capsule (EmC), which was suggested to be an important ventral pathway important for semantic processing [44,70]. The EmC was proposed to be an association fiber that projected to the frontal, parietal, and temporal cortices, and thus might be related to language processing [70,71,72]; however, given the trajectory of the EmC and the AF, a stroke lesion commonly caused co-occurring damage to these two fiber bundles and made it difficult to distinguish the linguistic effects of damage to the left LSAF and EmC in our patient sample. Third, we used a relatively rough measurement of language abilities, and did not further distinguish language dimensions (i.e., phonetics and semantics). Fourth, there is a disadvantage in the deterministic fiber-tracking algorithm regarding crossed fiber tracking, which may influence the reconstruction of the LSAF, UF, and IFOF. Finally, the UF is more difficult to measure successfully given the sensitivity of the diffusion sequence to artifacts around the sinuses [58], which can be observed in our patients with the heterogeneity of the UF reconstructed by DTI tractography. All these limitations need to be overcome with an enlarged sample and more advanced imaging technologies in future studies.

## 5. Conclusions

Exploring the neural substrate supporting language processing has been a longstanding topic in the field of brain neuroscience. Together with previous research, this work further expanded our knowledge about the left dual-stream frontotemporal network mediated by the LSAF and UF, and advanced our understanding of the linguistic roles of the left AF and UF. Our results suggested the potential importance of the structural integrity of the left LSAF rather than the UF in supporting residual language abilities in populations with acute/subacute PSA. Meanwhile, the current results did not suggest a functional interaction between the dorsal and ventral frontotemporal network mediated by the LSAF and UF, respectively. We discussed the possible explanations for these results, which are warranted to be further verified and generalized in the future.

## Figures and Tables

**Figure 1 brainsci-13-01324-f001:**
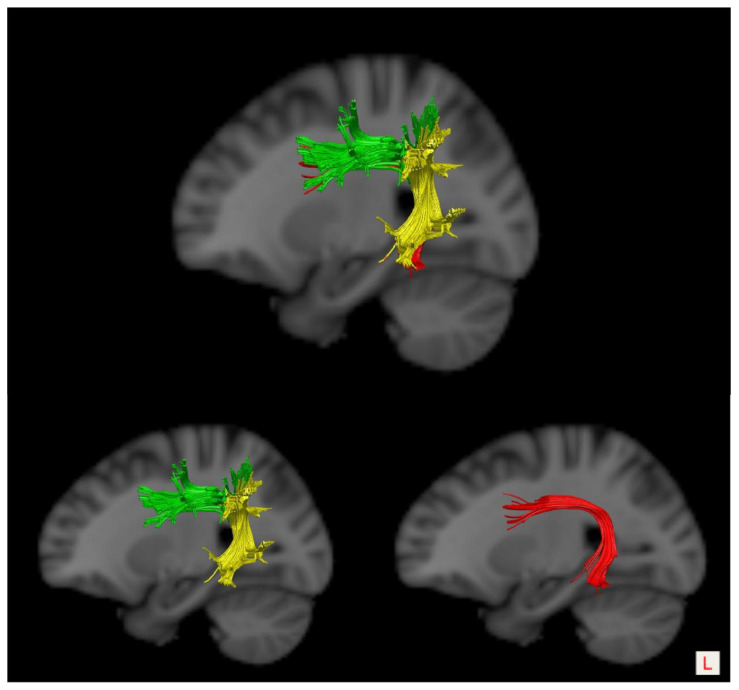
The virtual three-segment anatomical model of the AF: the green stream-anterior segment, the yellow stream-posterior segment, and the red stream-long segment. L, left.

**Figure 2 brainsci-13-01324-f002:**
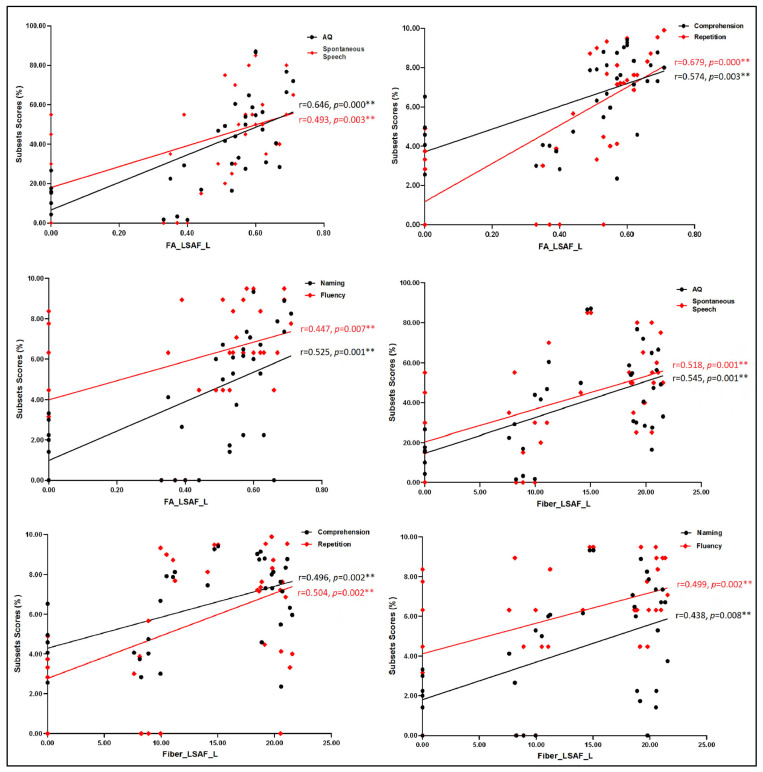
The partial correlation analyses between diffusion indices of the left LSAF/UF and language domain scores (controlled variable: lesion volume). The values, including FA, fiber, comprehension, repetition, naming, and fluency, were converted to a normal distribution by SQRT. **, *p* < 0.01.

**Figure 3 brainsci-13-01324-f003:**
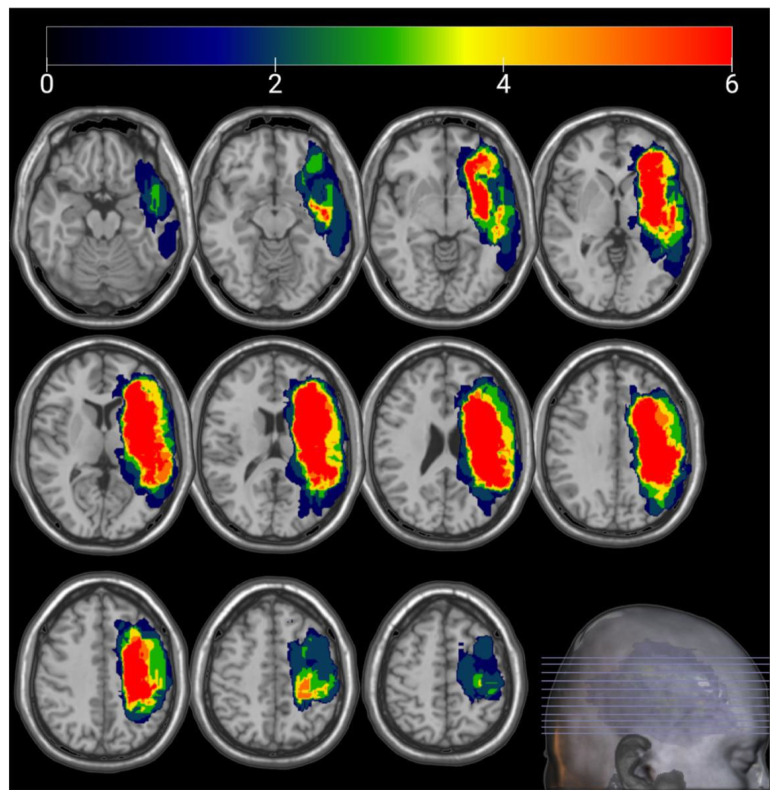
Lesion overlay map for all participants (N = 36). The hot color represents the regions with a high lesion overlap, while the cold color represents the regions with a lower lesion overlap.

**Figure 4 brainsci-13-01324-f004:**
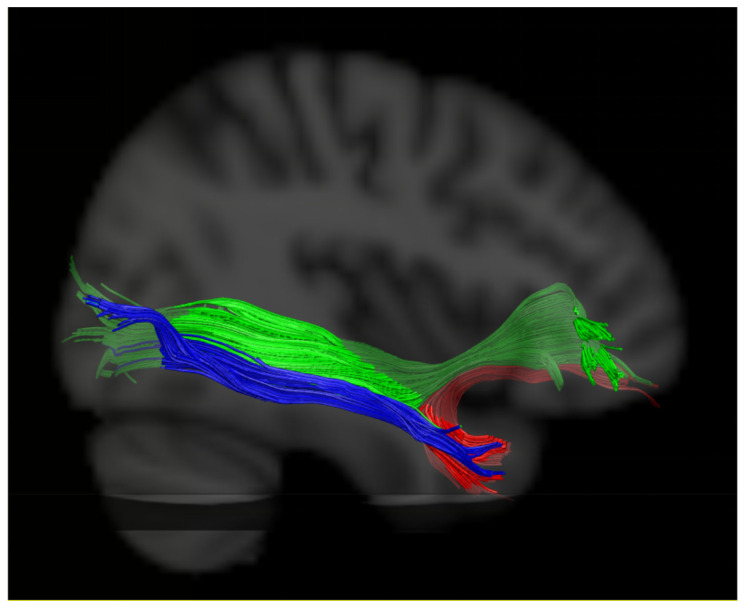
The proposed two ventral stream pathways supporting semantic language processing. IFOF—the green stream, ILF—the blue stream, and UF—the red stream.

**Figure 5 brainsci-13-01324-f005:**
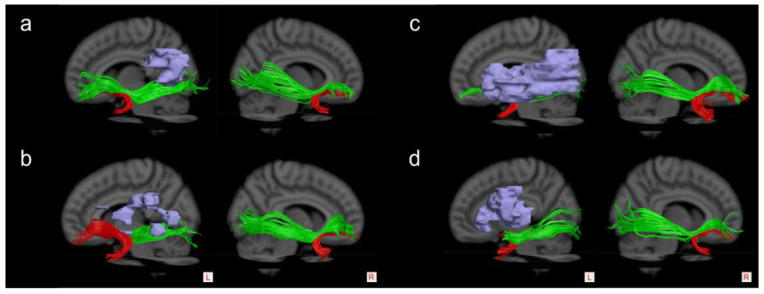
The reconstruction of the bilateral IFOF (the green stream) and UF (the red stream). (**a**), a sample of the intact IFOF and UF, 15 cases; (**b**), a representative of damage to the IFOF, 4 cases; (**c**), damage to the UF, 4 cases; and (**d**), damage to both the IFOF and UF, 13 cases. The light blue blocks are stroke lesions.

**Table 1 brainsci-13-01324-t001:** General demographic and clinical characteristics of all participants.

Patient ID	Sex/Age	Education (Years)	Time Post Onset (Days)	Stroke Type	Aphasia Type	Aphasia Severity	Lesion Site	Lesion Volume (cm^3^)
01	M/48	6	27	ICH	TMA	severe	Temporal lobe, basal ganglia, and corona radiate	22.18
02	F/59	6	12	Infarction	TSA	moderate	Basal ganglia, corona radiate	8.19
03	M/52	9	8	Infarction	TSA	mild	Basal ganglia, corona radiate, and centrum semiovale	7.94
04	F/56	12	18	ICH	Wernicke	severe	Temporal parietal lobe	68.07
05	M/64	9	8	Infarction	Global	very severe	Temporal and parietal lobe, insula	11.51
06	M/72	16	23	Infarction	Broca	severe	Basal ganglia, corona radiate	12.50
07	F/33	6	5	Infarction	Global	severe	Temporal and parietal lobe, insula	40.26
08	M/61	12	45	Infarction	Conduction	severe	Frontotemporal parietal lobe	21.02
09	M/59	12	17	ICH	Wernicke	severe	Temporal lobe, insula	75.34
10	M/49	15	10	Infarction	Broca	moderate	Basal ganglia, frontotemporal parietal lobe, and corona radiata	43.43
11	M/36	9	16	Infarction	Broca	moderate	Basal ganglia, temporal lobe, corona radiata, and centrum semiovale	27.79
12	F/56	9	81	ICH	Conduction	moderate	Basal ganglia, corona radiata	17.15
13	M/48	9	88	ICH	Broca	severe	Basal ganglia, frontotemporal lobe	29.30
14	M/38	15	83	ICH	Broca	moderate	Basal ganglia, frontal lobe	27.34
15	M/61	9	21	Infarction	Anomic	mild	Basal ganglia, frontal lobe	38.50
16	F/78	6	25	Infarction	Broca	severe	Basal ganglia, frontal lobe, corona radiate, and insula	23.62
17	M/42	16	64	ICH	Global	very severe	Basal ganglia, frontal lobe, and corona radiate	32.46
18	M/43	16	53	Infarction	Anomic	moderate	Frontoparietal lobe, insula	26.95
19	M/50	9	11	Infarction	Wernicke	severe	Temporo-occipital junction, insula	35.26
20	F/50	12	15	Infarction	Broca	severe	Temporal and parietal lobe, corona radiata, and insula	10.84
21	F/71	6	89	Infarction	Global	very severe	Frontotemporal parietal and occipital lobe	150.50
22	M/25	9	27	ICH	Anomic	mild	Temporo-occipital junction	21.82
23	M/53	16	56	ICH	TMA	severe	Frontotemporal parietal lobe, corona radiata, and insula	70.30
24	M/61	12	7	Infarction	Global	very severe	Temporoparietal-occipital junction, and insula	54.76
25	M/72	9	13	Infarction	Global	very severe	Temporoparietal-occipital junction, and corona radiata	40.78
26	M/62	9	23	Infarction	Global	very severe	Frontotemporal parietal lobe, basal ganglia, insula	41.53
27	M/64	6	13	ICH	Global	very severe	Frontotemporal parietal and occipital lobe	39.62
28	M/56	15	85	Infarction	Global	severe	Basal ganglia, frontoparietal lobe, and insula	76.59
29	M/34	12	77	Infarction	Global	severe	Frontotemporal parietal lobe	54.30
30	M/48	9	24	Infarction	Global	very severe	Frontotemporal parietal lobe, basal ganglia	1.44
31	M/48	15	10	ICH	Conduction	very severe	Frontotemporal and occipital lobe,	203.50
32	M/64	15	73	Infarction	MTA	severe	Frontoparietal lobe, insula	151.30
33	M/69	6	68	ICH	TMA	severe	Basal ganglia, frontal lobe	46.70
34	F/37	12	34	ICH	Anomic	mild	Basal ganglia, frontal lobe, corona radiata, and centrum semiovale	124.00
35	F/48	15	60	ICH	Global	very severe	Basal ganglia, corona radiate, and centrum semiovale	62.8
36	M/52	9	27	ICH	Global	very severe	Basal ganglia, corona radiate, and centrum semiovale	69.6

Note: F, female; ICH, intracerebral hemorrhage; M, male; MTA, mixed transcortical aphasia; TMA, transcortical motor aphasia; TSA, transcortical sensory aphasia.

**Table 2 brainsci-13-01324-t002:** The correlation analyses between language performance assessments and MRI measures.

	Lesion Volume		FA_LSAF_L		FA_LSAF_R		Fiber_LSAF_L		Fiber_LSAF_R		FA_UF_L		FA_UF_R		Fiber_UF_L		Fiber_UF_R	
	ρ	*p*	ρ	*p*	ρ	*p*	ρ	*p*	ρ	*p*	ρ	*p*	ρ	*p*	ρ	*p*	ρ	*p*
Aphasia quotient	−0.359 *	0.032	0.761 **	0.000	0.442 **	0.007	0.559 **	0.000	×	×	×	×	0.343 *	0.040	×	×	×	×
Spontaneous speech	×	×	0.588 **	0.000	×	×	0.508 **	0.002	×	×	×	×			×	×	×	×
Comprehension	×	×	0.637 **	0.000	0.476 **	0.003	0.440 **	0.007	×	×	×	×	0.343 *	0.040	×	×	×	×
Repetition	×	×	0.768 **	0.000	0.459 **	0.005	0.401 *	0.015	×	×	×	×	0.426 *	0.010	×	×	×	×
Naming	−0.351 *	0.036	0.639 **	0.000	0.401 **	0.015	0.450 **	0.006	×	×	×	×	0.361 *	0.030	×	×	×	×
Fluency	×	×	0.459 **	0.005	×	×	0.430 **	0.009	×	×	×	×			×	×	×	×

Note: FA_LSAF_L, means FA of the left LSAF; FA_LSAF_R, means FA of the right LSAF; Fiber_LSAF_L, fiber number of the left LSAF; Fiber_LSAF_R, fiber number of the right LSAF; FA_UF_L, mean FA of the left UF; FA_UF_R, mean FA of the right UF; Fiber_UF_L, fiber number of the left UF; Fiber_UF_R, fiber number of the right UF; *, *p* < 0.05; **, *p* < 0.01; ×, *p* > 0.05.

**Table 3 brainsci-13-01324-t003:** The multiple linear regression analysis (Enter method, N = 36).

*Y*	*X* ^#^	*B* (*SE*)	*Beta*	*t*	*p*	*F* Value	Adjusted *R*^2^
Aphasia quotient	FA_UF_L	3.646 (32.725)	0.024	0.111	0.912	5.056 ^a^	0.481
	FA_LSAF_L	150.828 (39.336)	0.968	3.834	0.001 *		
	UF × LSAF	−111.755 (121.240)	−0.307	−0.922	0.365		
	Lesion volume	0.162 (1.241)	0.019	0.131	0.897		
Spontaneous speech	FA_UF_L	−14.531 (40.613)	−0.092	−0.358	0.723	2.419 ^a^	0.245
	FA_LSAF_L	94.882 (48.816)	0.592	1.944	0.032 *		
	UF × LSAF	−0.279 (150.461)	−0.001	−0.002	0.999		
	Lesion volume	0.941 (1.541)	0.108	0.611	0.546		
Comprehension ^#^	FA_UF_L	−1.339 (3.328)	−0.096	−0.402	0.691	3.413 ^a^	0.355
	FA_LSAF_L	11.973 (4.000)	0.842	2.993	0.006 *		
	UF × LSAF	−6.292 (12.329)	−0.189	−0.510	0.614		
	Lesion volume	−0.012 (0.126)	−0.015	−0.094	0.926		
Repetition ^#^	FA_UF_L	−2.525 (4.348)	−0.119	−0.581	0.566	5.869 ^b^	0.527
	FA_LSAF_L	18.203 (5.226)	0.840	3.483	0.002 *		
	UF × LSAF	4.206 (16.107)	0.083	0.261	0.796		
	Lesion volume	0.308 (0.165)	0.262	1.870	0.072		
Naming ^#^	FA_UF_L	−3.333 (4.711)	−0.174	−0.707	0.485	3.011 ^a^	0.315
	FA_LSAF_L	12.910 (5.662)	0.661	2.280	0.031 *		
	UF × LSAF	−0.280 (17.453)	−0.006	−0.016	0.987		
	Lesion volume	0.003 (0.179)	0.002	0.015	0.988		
Fluency ^#^	FA_UF_L	−1.771 (4.894)	−0.099	−0.362	0.720	1.777	0.151
	FA_LSAF_L	8.279 (5.883)	0.455	1.407	0.041 *		
	UF × LSAF	6.085 (18.132)	0.143	0.336	0.740		
	Lesion volume	0.285 (0.186)	0.288	1.534	0.137		

Note: *Y*, dependent variables; *X*, independent variables; L, left; ^#^, values were converted to a normal distribution by SQRT; ^a^, *p* < 0.05; ^b^, *p* < 0.001; *, *p* < 0.05.

## Data Availability

All data used in this study are available from the corresponding author upon reasonable request.

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
