# Peer review of "Effects of Damage to the Integrity of the Left Dual-Stream Frontotemporal Network Mediated by the Arcuate Fasciculus and Uncinate Fasciculus on Acute/Subacute Post-Stroke Aphasia"

_brainsci, 2023, doi:10.3390/brainsci13091324_

Round 1
Reviewer 1 Report
This is a potentially interesting study on the association between the integrity of two major fiber tracts (arcuate and uncinate fasciculi in both hemispheres) in residual language function in acute/subacute right-handed stroke patients (<90 days post-stroke). A major issue in the results concerns the group analyses (patients were categorized into 4 groups according to the integrity of each [or both] tracts) because group size is extremely small (8-10 patients/group). These group sizes do not permit to permit multivariate analyses necessary to control for covariates (lesion size and potentially education and time post-injury; the fact that the four groups do not differ on the latter two variables at p<.05 does not preclude meaningful differences). So I would strongly suggest to omit these analyses.
Correlational results in the total sample are more defendable with the current sample size. In these analyses, however, important covariates should also be included (lesion size, age, education, time post-injury) in the context of linear multiple regression models. This approach would also allow testing of interactions (UF by LSAF FA or fiber number) in separate models (including mean-centered values of the original variables and their products).
Minor
The first paragraph of the Intro should be removed.
The aphasia severity and lesion site rows in Table 3 are blank
Tables 3-4: define significant pairwise comparisons on each variable
Table 2. Please spell out AQ
Minor linguistic editing required
Reviewer 2 Report
Dear Author,
Congratulations for your interesting article. It is very well structured and the conclusions are clearly underlined.
After reviewing this interesting article which presents the results of a well conducted study regarding the correlation between the integrity of the dual-stream fronto-temporal network, mediated by the arcuate fasciculus and the uncinate fasciculus in post-stroke aphasia.
The authors fully respected the article template, the methods were accurately chosen, they respected the ethical issues, the citations are well noted, and the conclusions are clear and well sustained.
The introduction provides precious information regarding the importance of white matter fiber pathways and their essential role in language formation. They used data from recent literature and focused on investigating the integrity of arcuate fasciculus and uncinate fasciculus.
After analysing the results of diffusion tensor imaging scan in 36 right-handed native Chinese patients with different types of aphasia secondary to acute/subacute stroke, they concluded that the integrity of the left long segment of the arcuate fasciculus is definitely more important than uncinate fasciculus, language assessment being significantly positively associated with diffusion indices and fiber number of the left long segment of arcuate fasciculus.
The main question proposed by the authors is if the alteration of the left dorsal and ventral frontotemporal network (mediated by the long segment of arcuate fasciculus and uncinate fasciculus) contributes to development of aphasia in acute/subacute stroke.
Taking into consideration the great interest for studying the importance of the arcuate fasciculus in the last 30 years, directly observed in different article databases (PubMed, Science Direct, Research Gate, etc.,), this original article tries to bring new information regarding the specific function of different parts of the arcuate fasciculus and their importance in language functions. What is more interesting in this study, is that the authors assessed language using Western Aphasia Battery Test – WAB Test – Aphasia Quotient, in patients with lesions situated in the left arcuate fasciculus and uncinate fasciculus, mapping and exactly measuring the number of altered fibers. Furthermore, they connected the obtained data and concluded that there is a demonstrated correlation between the integrity of left arcuate fasciculus and spontaneous speech, repetition and fluency, as other studies concluded before.
As authors mentioned, there are also some shortcomings of this study: the small number of patients in each group; another disadvantage would be that the authors did not evaluate other neural tract involved into ventral pathways, required for semantic processing, namely the extreme capsule, especially because stroke lesions commonly cause damage in both structures (arcuate fasciculus and extreme capsule), being hard to distinguish between the linguistic effects of each tract lesion. All these limitations might be used as advantages for further new studies.
There is a small inadvertence regarding the introduction: starting with the 35 row until 43 row, there are some additional information from the article editing recommendations that should be deleted.
Regarding the references, they were carefully selected, being updated to recent literature, and correlated to previous older information. The authors respected the chronological appearance of each reference index in the text. They also correctly integrated the information from mentioned references in their discussions.
Reviewer 3 Report
This is a quantitative, observational study with the aim of "to investigate the correlation between the integrity of the left dual-stream frontotemporal network mediated by the arcuate fasciculus (AF) and uncinate fasciculus (UF) and acute/subacute post-stroke aphasia (PSA)". To this purpose, the linguistic performance of 36 adult or elderly people with aphasia was evaluated by means of a standardized test, and diffusion measures of the tracts of interest were obtained. In addition to the correlation analysis between these measures, four subgroups, divided according to the integrity of the studied tracts, had their performance in the language test compared. The hypotheses raised were that damage to these tracts would compromise the performance in the language tasks, and the degree of impairment would be greater when both pathways were damaged.
I consider that the main issue of the study is that the correlation analysis would not be sufficient to answer the formulated research hypothesis, considering the very limitations already pointed out by the authors of the study. In this case, it would be necessary to adapt the research question and its hypothesis more appropriately to the design of the study carried out.
Still, I highlight below some points that need the attention of the authors:
Introduction:
1- The first paragraph of the introduction does not seem to be part of the text. The same occurs at the beginning of results.
2- Page 2, line 47: clarify what would be “anterior, posterior”. The sentence lacks cohesion.
3- Page 2, line 67: sentence without cohesion and coherence.
4- Page 2, lines 85-86: the idea of tracts with linguistic contributions does not sound appropriate. In the same way, there is a value judgment associated with the idea of who are interested in the subject. I suggest reviewing it.
5- Page 3, lines 111-113: would it be appropriate to justify the choice, what are the advantages of studying the white matter fiber pathways?
6- Page 3, line 122: Figure 1 is cited, with no call in the text. It would be important to clarify to the reader what it is about. The same is true of other tables and figures throughout the article.
Materials and Methods:
7- Page 5, section 2.2: specify which language test measures were analyzed. Example: fluency is only mentioned in results. In addition, inform the reader of the meaning of the measures: the higher, better or worse is the performance?
Results/Discussion:
8- The results of the correlation analysis were not interpreted as to the strength of the correlation. It would be important for this data to be discussed more specifically.
9- Table 3: there are two rows (variables) with no data.
1- Section 3.3: the data presented in this section were not discussed, so their relevance was not evident.
Round 2
Reviewer 1 Report
The authors have implemented all the suggested revisions. One final comment concerns the potential for colinearity between independent variables in the multiple regression models (i.e., high correlations between UF, AF measures and their interaction).
Several stylistic and grammatical edits are warranted throughout the text.
E.g Correlations between measurement of diffusion indices
corelations analyses
scores percentage of language subsets: language domain score
support for
long projected(?) fiber pathway
was suggested directly to connect--> was suggested that it directly connects..
linguistic sub-items,--> domains
score percentage--> percentage score
And several demographic and stroke-related
subsets scores of language measures-->language domain scores
moderately-strongly positive correlations
